# Adherence to Mediterranean Diet among Students from Primary and Middle School in the Province of Taranto, 2016–2018

**DOI:** 10.3390/ijerph17155437

**Published:** 2020-07-28

**Authors:** Guglielmo Bonaccorsi, Federica Furlan, Marisa Scocuzza, Chiara Lorini

**Affiliations:** 1Department of Health Science, University of Florence, 50134 Florence, Italy; guglielmo.bonaccorsi@unifi.it (G.B.); scocuzza.marisa@gmail.com (M.S.); chiara.lorini@unifi.it (C.L.); 2School of Specialization in Hygiene and Preventive Medicine, University of Florence, 50134 Florence, Italy

**Keywords:** KIDMED test, Mediterranean diet, adherence, nutritional status, school projects

## Abstract

The Mediterranean diet represents one of the healthiest dietary patterns, but nowadays it is increasingly being ignored in schools and by families. The aim of this study is to assess the adherence to the Mediterranean diet by pupils living in a small Southern Italian municipality, and whether it is a predictor of nutritional status.The degree of adherence to the Mediterranean diet, the socio-economic status and the nutritional status of 314 students (6–14 years) were tested during the 2016/2017 and 2017/2018 school years with the help of a questionnaire comprising the Mediterranean Diet Quality Index for Children and Adolescents (KIDMED) test. Multivariate logistic regression analysis was used to assess the predictive role of the KIDMED score and the other variables with respect to nutritional status. Adherence to the Mediterranean diet is high, medium and poor in, respectively, 24.8, 56.4 and 18.8% of students; it varies depending on gender and age, with females and older students showing higher values. In the multivariate logistic regression model, sex and KIDMED level are become significant predictors of nutritional status. This study highlights the need for intervention in the form of school projects—also involving families—to promote healthier eating habits in younger generations.

## 1. Introduction

The Mediterranean diet, recognized by UNESCO in 2010 as an Intangible Cultural Heritage of Humanity, is the traditional food model that characterizes the eating habits of populations overlooking the Mediterranean Basin, especially Greece, Spain and Southern Italy. 

The Mediterranean diet is based on a high intake of low glycemic index carbohydrates, unsaturated fat, fruit and vegetables, and on a low intake of meat and dairy products. It is also characterized by low alcohol consumption and a limited intake of sugar in the form of sweet snacks. The Mediterranean diet involves the consumption of local and seasonal products and is enriched also by social habits such as sharing of food, frugality and sobriety [1].

A series of studies carried out on Mediterranean populations have shown that such a diet coupled with daily exercise can prevent obesity and associated syndromes, and it can also counteract the onset of chronic cardiovascular diseases, tumors and typeII diabetes [2]. Among the parameters altered by incorrect dietetic behaviors are increased waist size, blood triglycerides, HDL (high density lipoprotein) cholesterol levels and glycaemia as well as arterial blood pressure, which, together, form so-called metabolic syndrome. Nowadays, this syndrome is also common in children [3,4], for whom adoption of an active lifestyle and appropriate dietary habits since the very beginning years of life can significantly reduce the risk of the onset of these chronic conditions. 

Unfortunately, among children and adolescents, recent decades have witnessed a shift from the Mediterranean diet to a “Westernized” one consisting of a higher intake of meat, sugars and saturated fats [5]. These components are commonly found in ready-to-eat meals and at fast food restaurants. In the Mediterranean countries, these habits are leading to an increase in the number of overweight people [6], followed by an increase in the number of subjects who will manifest cardiovascular diseases, type II diabetes and tumorsas adults. 

The reasons behind this shift are economic, social and cultural. In particular, economic globalization seems to play a fundamental role in the neglect of the consumption of local products, leading to a change in household eating habits [7], a decrease in the attention paid to food quality and quantity and, last but not least, to questionable dietary choices in schools (canteens and vending machines).

The education sector needs to develop—in terms of social accountability—projects aimed at raising awareness of healthy nutrition among pupils [8], so that the onset of chronic diseases typical of adulthood, and as an impact of social and economic elevation, can be prevented.

In order to implement these changes and be able to choose tailored projects in schools and households, it is necessary to estimate the percentage of children and adolescents who follow the Mediterranean diet [9]. This is achievable through specific measures, such as the Mediterranean Diet Quality Index for Children and Adolescents (KIDMED) test, originally conceived in Spain and now widespread in Europe, especially in the Mediterranean area [10].

The aim of this study is to assess the adherence to the Mediterranean diet by pupils attending primary schools, and whether it could be considered—along with physical and socioeconomic determinants—a predictor of nutritional status.

## 2. Materials and Methods 

### 2.1. Study Design and Participants

This study was carried out in Marina di Ginosa in the Ginosa municipality, in the province of Taranto, Southern Italy. Marina di Ginosa is a small seaside locality and sits in the middle of a geographic area well known for its Mediterranean diet. The sample consisted of 314 students aged between 6 and 14, which represented 95% of the total attending the IstitutoComprensivo ‘Raffaele Leone’ in Marina di Ginosa; the survey was carried out during the 2016/2017 and 2017/2018 school years. In the study, each participant was provided with a questionnaire that anonymously collected data on age, anthropometric parameters (i.e., weight and height) as well as parental qualifications. The KIDMED test was parallelly administered in order to evaluate the adherence of each participant to the Mediterranean diet. The Headmaster presented the project to the teachers via the school board and, once approved by the teachers, he sent a letter to inform the parents; the parent–teacher association (PTA) finally supported the project and the parents agreed unanimously. Older students attending secondary school were allowed to self-administer both the test and the questionnaire during school hours; the younger ones were allowed to fill in the test and the questionnaire at home, with the help of their parents.The answers were collected at school using a digital platform, called Kahoot, in order to preserve students’ privacy, ensuring thatthe participants’ answers were protected. With regard to the percentage of missing responses, 5% was lost among the younger students who filled the test and the questionnaire at home, because their parents did not hand back the responses. In the first school year (2016/2017), students attending primary school were enrolled, while in the second school year (2017/2018), those attending the secondary school were included.

### 2.2. KIDMED Test and Questionnaire

The Mediterranean Diet Quality Index for Children and Adolescents (KIDMED) test, developed by Serra-Majem et al., is a validated tool [11,12,13,14] to determine the level of adherence to the Mediterranean diet in individuals aged between 2 and 24 years old, which has also been validated in Italian language [15]. It is composed of 16 items modeled on simple ‘yes or no’ questions concerning daily eating habits and focuses on the consumption of different food groups (Table 1), with an associated score of −1, 0 or 1. The sums of the score for each answer produce an index that allows the classification of the subjects into one of the following levels of adherence to the Mediterranean diet: (1) >8, high; (2) 4–7, medium and (3) ≤3, poor. In addition to the KIDMED test, a questionnaire on gender, age and anthropometric data (weight in kilograms and height in centimeters) was also anonymously self-administered. The results of the questionnaire were gathered and processed in an Excel spreadsheet to obtain the Body Mass Index (BMI) of each participant, as a proxy of nutritional status. The BMI reference values were obtained from Cole et al., as suggested by the International Obesity Taskforce (IOTF) [16]. According to Cole BMI reference values, the students were classified as overweight, obese, normal weight or underweight. Additionally, the questionnaire required students to state the education level of their parents, which was meant to provide an indicator of the socio-economic status of the family unit to be associated with the health of each student. The questionnaire assessed whether the parents held a middle school, highschool or university degree; and also provided answer choices such as ‘I don’t know/I don’t want to answer’.

### 2.3. Data Analysis

Data are described as percentages or as the mean and standard deviation. The association between categorical variables was assessed using a chi-squared test, while, for continuous variables, Student’s t test or ANOVA were conducted. Multivariate logistic regression analysis was performed to calculate the association as the Odds Ratio (OR) of the nutritional status (outcome variable: ‘overweight or obese’ vs. ‘normal weight or underweight’) and the other variables (predictors: sex, age, educational level of the mother, educational level of the father, and levels of adherence to the Mediterranean diet). A multivariate logistic regression analysis was conducted using a backwards stepwise procedure. For all the analyses, an alpha level of 0.05 was considered significant. The analyses were conducted using IBM SPSS Statistics 25TM (IBM Corp., Armonk, NY, USA).

## 3. Results

### 3.1. Sample Description

In total, 314 students participated in this study. Among them, 150 were male (47.8%) and 164 were female (52.2%), with a mean age of 10 (2.4 SD). Anthropometrics (participants’ weight in kg, height in cm and BMI) reveal that 8% of the participants were obese and 26.4 % overweight, while 57.6 % had a normal weight and 8% were underweight.

### 3.2. KIDMED Test Responses and Adherence to Mediterranean Diet

The KIDMED test responses are shown in Table 2. In particular, even if a good proportion of teenagers indicate that they do consume raw or cooked vegetables, only a much smaller proportion do so more than once a day (27.4%). Fishery products are consumed in appropriate amounts (two or three times a week) by only half of the population under study (56.1%). On the other hand, more than half of the individuals regularly consume legumes (69.7%). The intake of pasta and rice is substantial among teenagers, and extravirgin olive oil is regularly preferred to butter in dressings by 90% of the population. Dairy products and cheese are consumed daily by about half the individuals.Unfortunately, onethird of the population states that they skip breakfast regularly, and among those who eat breakfast, a large majority generally consume pastry (31.5%). The consumption of pastries is associated with a high intake of refined carbohydrates, which contributes to a sudden rise in the levels of blood glucose and insulin. Fortunately, only a very small proportion of the population declare that they eat at fast-food restaurants more than once a week (16.2%); however, almost half of these individuals also declare that they consume sweets and candies more than once a day.

Statistical analyses revealed that adherence to the Mediterranean diet was high in 24.8%, medium in 56.4% and poor in the remaining 18.8% of the population under study. Regarding the KIDMED score, the average was 5.91 (Median: 6. SD: 2.395. Minimum: −1. Maximum: 11). Table 3 describes the adherence to the Mediterranean diet according to the other collected variables.

### 3.3. Determinants of the Adherence to the Mediterranean Diet

Age, gender, nutritional status and parents’ education were not significantly related to the adherence to the Mediterranean diet, although some tendencies emerged (Table 3).

The adherence to the Mediterranean diet tend to varydepending on the gender and age of participants, with younger females tending to show higher levels of adherence in relation to younger males, and older females and males both tending to show higher adherence in relation to younger females and males. Specifically, participants with poor adherence presented a mean age of 9.55 years (SD 2.41) while those with high adherence were older (mean age = 10.43 years, SD 2.154). The latter observation seems to suggest a relatively increased awareness of healthy feeding habits over time. Moreover, nutritional status tends to be related to adherence to the Mediterranean diet. With a *p* value (0.06) near to the alpha level, a higher prevalence of overweight and obesity was observed among those with lower adherence to the Mediterranean diet. 

The nutritional status, as indicated by the BMI, tends to belinked to gender. With a *p* value (0.06) near to the alpha level, the male subgroup showed a higher prevalence of overweight and obesity in relation to its female counterpart (Table 4). Moreover, by combining the data on the nutritional status and age of all participants, it is found that obese and underweight individuals are, on average, younger than overweight and normal-weight people (Table 4). The nutritional status was significantly associated with age and father’s education level. In the multivariate logistic regression model, sex and level of adherence to the Mediterranean diet, according to the KIDMED score, were confirmed to be significant predictors of nutritional status. Females presented an OR = 0.564 (*p* < 0.05) of being overweight or obese compared to males; and students with medium and poor adherence to the Mediterranean diet showed an OR of 2.314 and 2.776, respectively, of being overweight or obese compared to those with high adherence (Table 5).

## 4. Discussion

Several published studies carried out in Mediterranean countries, such as Greece, Spain and Cyprus, suggest that the adherence to the Mediterranean diet among teenagers is progressively and steadily dropping over time [17]. In particular, according to a recent review [18], in many countries where the adherence was previously high (Greece, Spain, Portugal, Italy and Turkey), the current levels—as resulting from the KIDMED test—are, at best, mediocre. Moreover, studies considering the Mediterranean diet score index rather than KIDMED [19] have shown comparable results in terms of adherence to the Mediterranean diet among teenagers. Similarly, our study reveals that the most representative KIDMED index category is that of average adherence.

Overall, we can confirm that the eating habits of this sample are divergent from those that were once typical of Mediterranean people, especially in Southern Italy [20].

Moreover, multivariate analysis shows that the level of adherence to Mediterranean diet significantly predicts, with sex, the nutritional status of the students: the higher the adherence, the lower the likelihood of being overweight or obese.

The causes behind this change are multiple and complex, and related to economic, cultural and social constraints. Considering that teenagers spend most of their time in school and at home, a lot remains to be done in these environments, especially since single-paycheck families and families in which parents have a low education level tend to overlook the quality and quantity of the food purchased. Concerning the variability among individuals, just as in other studies carried out in Spain, Greece and Italy, we can observe a positive correlation between the level of adherence to the Mediterranean diet and the female gender, as women tend to pay more attention to the type of food they consume [21].

Since children and teenagers spend most of their time in schools, it is appropriate to consider these places, together with household habits, as environmental determinants of health, nutritional status and the food choices of children.

Italy has, since 2007, adopted a National Surveillance System, ‘Okkioalla salute’, which helps estimate childhood obesity and put in place preventive measures to counteract this phenomenon. Okkioalla salute is linked to the WHO’s Childhood Obesity Surveillance Initiative (COSI), which is an excellent example of an evidence-based approach and synergy between institutions. Indeed, the investigations are carried out by several institutions: the health sector (Ministry of Health), several individual regional authorities, the Ministry of Education, and different universities and schools. The latest report, dated 2016 [22], highlighted that three out of four schools scheduled regular nutritional education courses taught by school teachers; more than 40% of schools launched initiatives to promote healthy eating habits, and more than 30% of schools organized physical activity events that encouraged the participation of parents. In more than 50% of schools, children take part in at least two hours of physical activity per week, and in more than 60% of schools, more opportunities for physical exercise are available to pupils.

Moreover, in Italy, 50.9% of schools provide healthy food such as yogurt, fruit or milk (both for breakfast and for morning or afternoon breaks; 9.8% of schools provide vending machines with food accessible to children (this percentage has been decreasing since 2014). In 2016, 70.8% of primary schools, together with other institutions and associations, put in place or took part in initiatives to promote healthy eating habits. Moreover, 31.6% of schools initiated nutritional education programs in collaboration with the local health authority; 41.9% of schools created initiatives, which included the participation of families, to promote healthy eating habits. In 2016, two requests were forwarded to use iodized salt in canteens and to promote initiatives to encourage a reduced consumption of salt and/or in the substitution of normal salt with iodized salt.

Apart from Italy, a large number of school interventions are being organized all over Europe. For example, mention can be made of the ‘school fruit, vegetables and milk scheme’ that has been distributing fruit, vegetables and milk in schools across the European Union since August 2017 [23]; and it is part of a wider program of education about European agriculture and the benefits of healthy eating. It promotes a healthy diet through a list of products, which children receive following the approval of national health and nutrition administrations. In the list, priority is given to fresh fruit, vegetables, and milk and the choice is based on seasonality, variety and availability. Moreover, no added sugars, salt, fat and sweeteners or artificial flavors are allowed.

To quote another example, ‘Ensemble Prévenonsl’Obésité Des Enfants’, meaning Together Let’s Prevent Childhood Obesity (EPODE) [24] is a coordinated, capacity-building approach aimed at reducing childhood obesity through a societal process in which local environments, childhood settings and family norms are encouraged to promote the adoption of healthy lifestyles in children aged between 0 and 12. EPODE was first launched in 2004 in 10 French pilot communities and has since expanded to more than 500 communities worldwide. The EPODE methodology promotes the involvement of multiple stakeholders at the central level (ministries, health groups, non-governmental organizations and private partners) and at a local level (political leaders, health professionals, families, teachers, local NGOs (non-governmental organization) and the local business community) through the adoption of long-term programs and initiatives to promote healthy lifestyles in school and families in a sustainable manner.

Similarly, in Greece, the PAIDEIATROFI program [25], a community-based intervention program, aims to promote a lifestyle change among children to fight increasing obesity in families and schools. The municipality is the center where the majority of daily activities are developed and where the family gets in touch with a variety of social entities such as school, work, health, transportation and sports clubs.

The strength of this study is based on the evaluation of the level of adherence to the Mediterranean diet via the KIDMED test, a scientifically validated tool adopted in several studies carried out in the Mediterranean area, enabling a comparison with the situation in other countries.

One of the limitations of the study is the fact that we relied on anthropometric values declared by the individuals themselves, rather than on parameters measured independently. This might have introduced a bias in the evaluation of the association between adherence to the Mediterranean diet and BMI values.

## 5. Conclusions

In conclusion, this study highlights the need for intervention in the form of projects aimed at improving the quality of lifestyle and at promoting healthier eating habits among the younger generations. This will help prevent an early onset of metabolic diseases as well as obesity, and will also help educate families on the adoption of a healthy lifestyle, which will be beneficial for young individuals throughout their life.

## Figures and Tables

**Table 1 ijerph-17-05437-t001:** Mediterranean Diet Quality Index for Children and Adolescents (KIDMED) test to assess Mediterranean diet adherence [10].

**KIDMED TEST**	**SCORING**
Takes a fruit or fruit juice every day	1
Has a second fruit every day	1
Has fresh or cooked vegetables regularly once a day	1
Has fresh or cooked vegetables more than once a day	1
Consumes fish regularly (at least 2–3/week)	1
Goes >1/week to a fast food restaurant?	−1
Likes pulses and eats them >1/week	1
Consumes pasta or rice almost every day (5 or more per week)	1
Has cereals or grains (bread, etc.) for breakfast	1
Consumes nuts regularly (at least 2–3/week)	1
Uses olive oil at home	1
Skips breakfast	−1
Has a dairy product for breakfast (yogurt, milk…)	1
Has commercially baked goods or pastries for breakfast	−1
Takes two yogurts and/or some cheese (40 g) daily	1
Takes sweets and candy several times every day	−1
**KIDMED INDEX**	**ADHERENCE TO MEDITERRANEANDIET**
Score ≤ 3 points	Poor
Score 4–7 points	Medium
Score≥ 8 points	High

**Table 2 ijerph-17-05437-t002:** Answers to each item of KIDMED questionnaire.

Item	Yes N (%)	No N (%)
Takes a fruit or fruit juice every day	248 (79)	66 (21)
Has a second fruit every day	169 (53.8)	145(46.2)
Has fresh or cooked vegetables regularly once a day	150 (47.2)	164(52.2)
Has fresh or cooked vegetables more than once a day	86 (27.4)	228 (72.6)
Consumes fish regularly (at least 2–3/week)	176 (56.1)	148 (43.9)
Goes >1/week to a fast food restaurant?	263 (83.8)	51 (16.2)
Likes pulses and eats them >1/week	219 (69.7)	95 (30.3)
Consumes pasta or rice almost every day (5 or more per week)	259 (82.5)	55 (17.5)
Has cereals or grains (bread, etc.) for breakfast	188 (59.9)	126 (40.1)
Consumes nuts regularly (at least 2–3/week)	161 (51.3)	153 (48.7)
Uses olive oil at home	292 (93)	22 (7)
Skips breakfast	222 (70.7)	92 (29.3)
Has a dairy product for breakfast (yogurt, milk…)	250 (79.6)	64 (20.4)
Has commercially baked goods or pastries for breakfast	99 (31.5)	215 (68.5)
Takes two yogurts and/or some cheese (40 g) daily	140 (44.6)	174 (55.4)
Takes sweets and candy several times every day	190 (60.5)	124 (39.5)

**Table 3 ijerph-17-05437-t003:** Adherence to Mediterranean diet according to KIDMED questionnaire on sex, nutritional status, mother and father’s education level.

		Adherence to Mediterranean Diet According to KIDMED			Total
		POOR N (%)	MEDIUM N (%)	HIGH N (%)	
**Age, mean (*p* =0.09)**		9.55	10.22	10.43	
**Sex (*p* =0.19)**	Females	25 (15.2)	94 (57.3)	45 (27.4)	164 (100)
	Males	34 (22.7)	83 (55.3)	33 (22.0)	150 (100)
**Nutritional status** **(*p* = 0.06)**	Normal weight	30 (16.6)	99 (54.7)	52 (28.7)	181 (100)
	Obese	8 (32.0)	15 (60.0)	2 (8.0)	25 (100)
	Underweight	2 (8.0)	14 (56.0)	9 (36.0)	25 (100)
	Overweight	19 (22.9)	49 (59.0)	15 (18.1)	83 (100)
**Mother’s education level** **(*p* = 0.1)**	Graduation	31 (19.7)	92 (58.6)	34 (21.7)	157 (100)
	Degree	7 (14.6)	23 (47.9)	18 (37.5)	48 (100)
	Junior high school	15 (19.2)	49 (62.8)	14 (17.9)	78 (100)
	I don’t know/I don’t want to answer	6 (19.4)	13 (41.9)	12 (38.7)	31 (100)
**Father’s education level** **(*p* = 0.06)**	Graduation	22 (15.8)	87 (62.6)	30 (21.6)	139 (100)
	Degree	6 (15.8)	16 (42.1)	16 (42.1)	38 (100)
	Junior high school	23 (22.3)	59 (57.3)	21 (20.4)	103 (100)
	I don’t know/I don’t want to answer	8 (23.5)	15 (44.1)	11 (32.4)	34 (100)

**Table 4 ijerph-17-05437-t004:** Gender and age by nutritional status.

		Nutritional Status N (%)				Total
		Underweight 25 (8)	Normal Weight 181 (57.6)	Overweight 83 (26.4)	Obese 25 (8)	
**Sex (*p* = 0.06)**	Females	16 (9.8)	102 (62.2)	37 (22.6)	9 (5.5)	164 (100)
	Males	9 (6.0)	79 (52.7)	46 (30.7)	16 (10.7)	150 (100)
**Age (*p* = 0.02)mean ± SD**		9.48 ± 2.9	10.4 ± 2.5	9.94 ± 2.1	9.04 ± 2.3	10 ± 2.4
**Mother’s education level (*p* = 0.16)**	Graduation	12 (7.6)	87 (55.4)	47 (29.9)	11 (7.0)	157 (100)
	Degree	6 (12.5)	27 (56.2)	8 (16.7)	7 (14.6)	48 (100)
	Junior high school	7 (9.0)	43 (55.1)	22 (28.2)	6 (7.7)	78 (100)
	I don’t know/I don’t want to answer	0 (0)	24 (77.4)	6 (19.3)	1 (3.3)	31 (100)
**Father’s education level (*p* = 0.01)**	Graduation	15 (10.8)	75 (54)	44 (31,6)	5 (3.6)	139 (100)
	Degree	4 (10.5)	23 (60.5)	7 (185)	4 (10.5)	38 (100)
	Junior high school	6 (5.8)	58 (56.3)	24 (23.3)	15 (14.6)	103 (100)
	I don’t know/I don’t want to answer	0 (0)	25 (73.5)	8 (23.5)	1 (3)	34 (100)

**Table 5 ijerph-17-05437-t005:** Multivariate logistic regression model. Outcome variable: ‘overweight or obese’ vs. ‘normal weight or underweight’. Sign.= significance; S.E. = standard error.

		OR	Sign.	S.E.
**Sex**	Males	1	-	-
	Females	0.564	0.018	0.243
**KIDMED** **level**	High	1	-	-
	Medium	2.314	0.047	0.422
	Poor	2.776	0.029	0.467

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
