# Peer review of "Adherence to Mediterranean Diet among Students from Primary and Middle School in the Province of Taranto, 2016–2018"

_ijerph, 2020, doi:10.3390/ijerph17155437_

Round 1

Reviewer 1 Report

Since 2004 the KIDMED questionnaire has been used to evaluate adherence to the mediterranean diet but the last few decades have seen considerable changes in the dietary habits of the Mediterranean population, especially in children and adolescents. High energy intake and the massive consumption of high-sugar foods as fruit juice, refined cereals or refined grains in the MD may have negative effects on health. there was a paradigm shift about the daily consumption of fruit juice and whole grains.

It must consider the  responsibility for  use the updating KIDMED tool used to assess adherence to the MD. These changes have led to an update of the KIDMED questionnaire in English. (Cesare Altavilla and Pablo Caballero-Pérez; An update of the KIDMED questionnaire, a Mediterranean Diet Quality Index in children and adolescents. Public Health Nutrition: 22(14), 2543–2547). they propose some modifications to the KIDMED questionnaire to provide a tool according to the new recommendations for a healthy diet in children and adolescents.

Author Response

Point 1

It must consider the  responsibility for  use the updating KIDMED tool used to assess adherence to the MD. These changes have led to an update of the KIDMED questionnaire in English. (Cesare Altavilla and Pablo Caballero-Pérez; An update of the KIDMED questionnaire, a Mediterranean Diet Quality Index in children and adolescents. Public Health Nutrition: 22(14), 2543–2547). they propose some modifications to the KIDMED questionnaire to provide a tool according to the new recommendations for a healthy diet in children and adolescents.

Response 1: Thanks for the suggestion. Our survey has been conducted in 2016-2018, so at that time the reviewed version of the KIDMED was not already published. We will consider your suggestion for future studies.

Reviewer 2 Report

The study addresses an interesting and current topic. The manuscript seems well written to me. It is a very interesting study, but it needs some improvements.

The authors should add in the methodology section the ethical considerations that have been followed in the study. Participants are minors.

It would be appropriate to describe whether the KIDMED test has been validated on Italian students.

The procedure is explained very briefly. It would be interesting to know how the teachers have participated in the study and how the data has been collected in two academic years, as well as if there was informed consent from the adults. The authors do not clarify the information on the procedure.

The results values do not coincide in the text and in Table 3. Example "Specifically, the participants with low adherence had a mean age of 9.55 years (SD 2.41) 140 while those with high adherence were significantly older (mean age = 10.43 years), DE 2.154) "

The authors indicate that nutritional status was directly related to adherence to the Mediterranean diet despite not obtaining significant data. Likewise, without reaching significant data, it was described that the nutritional status, as indicated by the BMI index, was related to gender. It would be appropriate for them to clarify the meaning of these relationships / links.

Finally, I suggest that the authors provide more recent bibliographic citations both in the introductory section and in the manuscript discussion, to better understand the current situation of evaluating adherence to the Mediterranean diet.

Author Response

Point 1

The study addresses an interesting and current topic. The manuscript seems well written to me. It is a very interesting study, but it needs some improvements.

The authors should add in the methodology section the ethical considerations that have been followed in the study. Participants are minors.

Response 1: this information has been added in the methods, as follow: “The Headmaster presented the project to the teachers during the school board, and once approved by the teachers he sent a letter to inform the parents; the PTA (parent-teacher association) finally supported the project and the parents agreed unanimously.” (lines 76-78).

Point 2

It would be appropriate to describe whether the KIDMED test has been validated on Italian students.

Response 2: The KIDMED has been validated in Italian language in a study conducted in Tuscany and published in 2014. The following reference has been added, with a comment in the main text (lines 91,92).

Santomauro, F.; Lorini, C.; Tanini, T.; Indiani,  L.;  Lastrucci, V.;  Comodo, N.;  Bonaccorsi, G. Adherence to Mediterranean diet in a sample of Tuscan adolescents.Nutrition2014, 30,1379-83. doi: 10.1016/j.nut.2014.04.008.

Point 3

The procedure is explained very briefly. It would be interesting to know how the teachers have participated in the study and how the data has been collected in two academic years, as well as if there was informed consent from the adults. The authors do not clarify the information on the procedure.

Response 3: The methods section has been improved, according also to the first comment, as follow: The Headmaster presented the project to the teachers during the school board, and once approved by the teachers he sent a letter to inform the parents; the PTA (parent-teacher association) finally supported the project and the parents agreed unanimously. Older students, attending secondary school, were allowed to self-administer both the test and the questionnaire during school hours; the younger ones were allowed to fill in the test and the questionnaire at home , with the help of their parents. The answers were collected at school using a digital platform, called KAHOOT, in order to keep students’s privacy, avoiding that anyone could know the participants’ answers. With regard to the percentage of missing responses, a 5% has been lost among the younger who filled the test and the questionnaire at home, because their parents did not hand back the responses. In the first school year (2016/2017), students attending primary school were enrolled while in the second school year (2017/2018) those attending the secondary school were included.” (lines 76-87).

 Point 4 and 5

The results values do not coincide in the text and in Table 3. Example "Specifically, the participants with low adherence had a mean age of 9.55 years (SD 2.41) 140 while those with high adherence were significantly older (mean age = 10.43 years), DE 2.154) "

The authors indicate that nutritional status was directly related to adherence to the Mediterranean diet despite not obtaining significant data. Likewise, without reaching significant data, it was described that the nutritional status, as indicated by the BMI index, was related to gender. It would be appropriate for them to clarify the meaning of these relationships / links.

Responses 4 and 5: Thanks for the comments. We have modified this part of the results as follow: “Age, gender, nutritional status and parents’ education were not significantly related to the adherence to Mediterranean Diet, although some tendency emerged (Table 3).

The adherence to the Mediterranean diet tends to vary depending on gender and age of participants, with younger females tending to show higher levels of adherence in relation to younger males, and older females and males both tending to show higher adherence in relation to younger females and males. Specifically, participants with poor adherence presented a mean age of 9.55 years (SD 2.41) while those with high adherence are older (mean age=10.43 years, SD 2.154). The latter observation seems to suggest a relatively increased awareness of healthy feeding habits over time. Moreover, the nutritional status tend to be related to the adherence to the Mediterranean diet. With a p value (0.06) near to the alfa level, a higher prevalence of overweight and obesity was observed among those with lower adherence to the Mediterranean diet.

The nutritional status, as indicated by the BMI, tends to be linked to gender. With a p value (0.06) near to the alfa level, the male subgroup showed a higher prevalence of overweight and obesity in relation to its female counterpart (Table 4).” (lines 148-163).

Point 6

Finally, I suggest that the authors provide more recent bibliographic citations both in the introductory section and in the manuscript discussion, to better understand the current situation of evaluating adherence to the Mediterranean diet.

Response 6: The reference section has been updated with the following citations: 1,2,4,7,17,19.

Reviewer 3 Report

This is a useful study, but to have real impact should be much stronger.

References to the KIDMED tool should be given at first mention, not later. The reference cited (ref. 10) for the KIDMED tool is not a true validation, but rather a proof in use. Thus the construct of the study relies too heavily on one not fully validated tool.

The authors do not give details of how the data were collected - student recruitment, participation rates, whether the data were anonymised, whether they were collected in a setting where each child kept their answers confidential from other students, etc..

Parental education is indeed an important determinant of child nutrition and child behaviour. But the study would also benefit from assessing other demographic details such as indicators of poverty, poor housing, large family size, single adult households etc., mobility. No information is given of the socio-economic profile of the study venue town.

The study would have had much more impact if it had been conducted in more than one location.

Author Response

Point 1

References to the KIDMED tool should be given at first mention, not later. The reference cited (ref. 10) for the KIDMED tool is not a true validation, but rather a proof in use. Thus the construct of the study relies too heavily on one not fully validated tool.

Response 1:The reference for KIDMED has been brought forward (line 62). The reference you refer to (n. 10) is the pillar development study of this tool, to which the other researches refer, as indicated in the references we have added (line 90). Moreover, we have also cited the validation study of KIDMED in Italian language (Santomauro, 2014)* (lines 91,92).

Moreover, if useful to add in this study some statistical analysis in order to assess the validity of the Italian version of KIDMED we have used, we can perform them.

*Santomauro, F.; Lorini, C.; Tanini, T.; Indiani,  L.;  Lastrucci, V.;  Comodo, N.;  Bonaccorsi, G. Adherence to Mediterranean diet in a sample of Tuscan adolescents.Nutrition2014, 30,1379-83. doi: 10.1016/j.nut.2014.04.008.

Point 2

The authors do not give details of how the data were collected - student recruitment, participation rates, whether the data were anonymised, whether they were collected in a setting where each child kept their answers confidential from other students, etc..

Response 2: The methods section has been improved, according also to the comments of the second Referee, as follow: “In the study, each participant was provided with a questionnaire that anonymously collected data on age, anthropometric parameters (i.e. weight and height) as well as parental qualifications. The KIDMED test was parallely administered in order to evaluate the adherence of each participant to the Mediterranean Diet.

The Headmaster presented the project to the teachers during the school board, and once approved by the teachers he sent a letter to inform the parents; the PTA (parent-teacher association) finally supported the project and the parents agreed unanimously. Older students, attending secondary school, were allowed to self-administer both the test and the questionnaire during school hours; the younger ones were allowed to fill in the test and the questionnaire at home , with the help of their parents. The answers were collected at school using a digital platform, called KAHOOT, in order to keep students’s privacy, avoiding that anyone could know the participants’ answers. With regard to the percentage of missing responses, a 5% has been lost among the younger who filled the test and the questionnaire at home, because their parents did not hand back the responses. In the first school year (2016/2017), students attending primary school were enrolled while in the second school year (2017/2018) those attending the secondary school were included.” (lines 73-87).

Point 3

Parental education is indeed an important determinant of child nutrition and child behaviour. But the study would also benefit from assessing other demographic details such as indicators of poverty, poor housing, large family size, single adult households etc., mobility. No information is given of the socio-economic profile of the study venue town.

Response 3: Unfortunately, those information were not collected due to privacy issues. Future studies will take into account also this aspect.

 Point 4

The study would have had much more impact if it had been conducted in more than one location.

Response 4: Thanks for the comment. We really know that this is a local study, that could be more interesting in case it could include more schools from other geographic areas. The survey has been conducted only in a small town due to the interest shown by local stakeholders in nutritional education of children and adolescents, so it has to be considered a convenience sample. In the future, we will try to improve our study including other schools in a wider geographical area and assessing more information.

Round 2

Reviewer 1 Report

I understand, we have also used these tools, although they are not correct. It is difficult to find an index that indicates how correct your diet is, there are many factors involved. but we must update ourselves day by day and correct the flaws found for future investigations.

Reviewer 3 Report

Thank you for the actions taken to improve this manuscript.